# Improving Variational Autoencoder with Deep Feature Consistent and Generative Adversarial Training

**Xianxu Hou, [1,2] Guoping Qiu [1,2]**
[1] University of Nottingham, Nottingham, UK
[2] Shenzhen University, Shenzhen, China
`xianxu.hou@nottingham.edu.cn`
`guoping.qiu@nottingham.ac.uk`

## Abstract

We present a new method for improving the performances of variational autoencoder (VAE). In addition to enforcing the deep feature consistency principle thus ensuring the VAE output and its corresponding input images to have similar deep features, we also implement an adversarial generative training mechanism to force the VAE to output realistic and natural images. We present experimental results to show that the VAE trained with our new method outperform state of the art in generating face images with much clearer and more natural noses, eyes, teeth, hair textures as well as reasonable backgrounds. We also show that the VAE trained with the new method can extract more effective features that outperform state of the art in facial attribute recognition.

## 1 Introduction

Generative models, as a branch of unsupervised learning technique in machine learning, have become an area of active research in recent years. A generative model trained with a given image database can be useful in several ways. One is to learn the essence of a dataset and generate realistic images similar to those in the dataset from random vectors.The other is to learn reusable feature representations from unlabeled image datasets for a variety of supervised learning tasks such as image classification. In this paper, we propose a new method to train the variational autoencoder (VAE) (Kingma & Welling, 2013) to improve its performances in the aforementioned two applications. First, we use a deep feature consistent principle to ensure that the output image of the VAE to have deep features that are consistent with those of the input, and this is called DFC-VAE (Hou et al., 2017). Second, we use the principle of generative adversarial network (GAN) (Goodfellow et al. (2014)) to enforce the VAE output to resemble natural real images. We introduce several techniques to improve convergence of GAN training in this context. We present experimental results to show that our new method can generate face images with much clearer facial features such as eyes, nose, mouth, teeth, ears and hairs. We also show that the VAE trained by our method can extract much more effective features that outperform state of the art in facial attribute recognition.

## 2 Related Work

Several methods have been proposed to improve the performance of VAE. Kingma et al. (2014) proposes to build variational autoencoders by conditioning on either class labels or on a variety of visual attributes. Ridgeway et al. (2015) and Hou et al. (2017) consider replacing per-pixel loss with perceptual similarities using either multi-scale structural similarity score or a perceptual loss based on deep features extracted from pretrained deep networks. In addition, several recent papers (Denton et al., 2015; Radford et al., 2015; Im et al., 2016; Salimans et al., 2016; Chen et al., 2016; Arjovsky et al., 2017) have focused on improving the perceptual quality of the output of GAN and the training stability of GAN through architectural innovation and new training techniques. Our model combines the advantages of deep feature consistent VAE (DFC-VAE) (Hou et al., 2017) and Wasserstein GAN (WGAN) (Arjovsky et al., 2017) to improve the variational autoencoder.

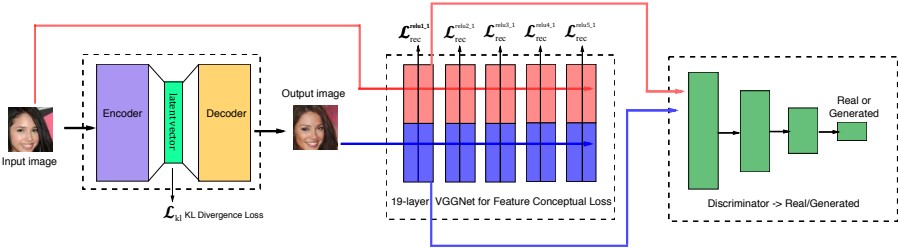

Figure 1: Model overview. From left to right: The Variational autoencoder (VAE), the VGGNet used for computing perceptual loss to enforce deep feature consistency between the VAE's output and the input images, the discriminator trained based on the principle of WGAN. Note that the inputs to the discriminator come from the first convolutional layer of the VGGNet.

## 3 DEEP FEATURE CONSISTENCY, VAE AND WGAN

As shown in Figure 1, our model consists of three components: a variational autoencoder, a pre-trained VGGNet (Simonyan & Zisserman, 2014) for feature extraction and a classifier network used as discriminator. It is used to improve the VAE using the deep features extracted by the VGGNet and use the principle of GAN to enforce the VAE to output natural and realistic images. Both the encoder and the decoder are deep residual convolutional neural networks with a 100-dimensional latent vector. The encoder compresses the input image into the latent feature which is decoded to an output image. The VGGNet is used to extract deep features to construct the perceptual loss. The VAE also serves as a generator and works with discriminator to perform the GAN game. Instead of feeding the pixels to the discriminator, we use the first layer output of the VGGNet as the input of the discriminator and found that this can significantly improve the stability of the GAN training.

Similar to Johnson et al. (2016), our feature reconstruction loss is also defined as the Euclidean distance between the feature maps of a reconstruction image and a reference image. Instead of only using a single layer features, we leverage visual features at multiple scales and use the outputs of the five convolutional layers of the VGGNet, i.e., $L_{rec} = \sum_{i=1}^{5} \frac{100}{C_i^2} L_i$, where $L_i$ and $C_i$ are the feature loss and the number of filters at $i^{th}$ layer, respectively.

In order to improve the stability of WGAN training (Arjovsky et al., 2017), unlike traditional GAN that directly feeds the raw real images and the generated images to a discriminator, we first extract the first layer features of the pretrained VGGNet and feed them to the discriminator network. Another technique is to further relax the constraint on the output of the discriminator network. WGAN proposes to remove the last Sigmoid layer in the generator and use 1 and -1 as ground-truth for real and generated images. In our experiments, we found that GAN training could collapse and the VAE training tends to dominate when using the default setting of WGAN. Instead, by simply using bigger values to represent ground-truth (10 and -10), we can achieve a good balance between the VAE and GAN training, producing better results.

## 4 EXPERIMENT

**Image generation.** Our model is evaluated on the CelebA dataset (Liu et al., 2015). Figure 2 (a) gives some qualitative examples of the generated images by DCGAN (Radford et al., 2015), DFC-VAE (Hou et al., 2017) and our new VAE-WGAN from random vectors. We can see that our new method can generate more consistent and realistic human faces with much clearer noses, eyes, teeth, hair textures as well as reasonable background. Furthermore, we have conducted experiments to manipulate the facial attributes in the learned latent space. For a given attribute like *smiling*, 2,000 smiling face samples are fed into the trained encoder to generate 2000 latent vectors. The average of these 2000 latent features forms latent feature $z_{smiling+}$. Similarly, we use 2000 non smiling face samples to generate a non-smiling latent vector $z_{smiling-}$. Finally the difference $z_{smiling} = z_{smile+} - z_{smile-}$, which in effect takes away any non-smiling attributes from the smiling latent feature, is used as the semantic representation for the attribute *smiling*. Similarly, we use the same approach to constructing other semantic attribute latent features for *Bald*, *Black hair*, *Eyeglass*,

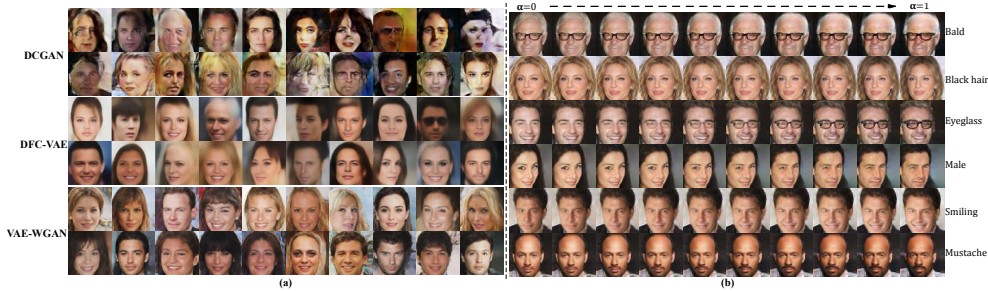

Figure 2: Qualitative results. (a) shows the faces generated from random vectors. (b) shows the results for facial attributes manipulation in the latent space, e.g. $z = z + \alpha z_{smiling}$.

Table 1: Performance comparison of 40 facial attributes prediction.

| Method | 5 Shadow | Arch. Eyebrows | Attractive | Bags Un. Eyes | Bald | Bangs | Big Lips | Big Nose | Black Hair | Blond Hair | Blurry | Brown Hair | Bushy Eyebrows | Chubby | Double Chin | Eyeglasses | Goatee | Gray Hair | Heavy Makeup | H. Cheekbones | Male |
|---|---|---|---|---|---|---|---|---|---|---|---|---|---|---|---|---|---|---|---|---|---|
| FaceTracer (Kumar et al., 2008) | 85 | 76 | 78 | 76 | 89 | 88 | 64 | 74 | 70 | 80 | 81 | 60 | 80 | 86 | 88 | 98 | 93 | 90 | 85 | 84 | 91 |
| PANDA-w (Zhang et al., 2014) | 82 | 73 | 77 | 71 | 92 | 89 | 61 | 70 | 74 | 81 | 77 | 69 | 76 | 82 | 85 | 94 | 86 | 88 | 84 | 80 | 93 |
| PANDA-l (Zhang et al., 2014) | 88 | 78 | **81** | 79 | 96 | 92 | 67 | 75 | 85 | 93 | 86 | 77 | 86 | 86 | 88 | 98 | 93 | 94 | **90** | 86 | 97 |
| LNets+ANet (Liu et al., 2015) | **91** | 79 | **81** | 79 | **98** | **95** | 68 | 78 | **88** | **95** | 84 | 80 | **90** | 91 | 92 | **99** | **95** | **97** | **90** | **87** | **98** |
| VAE-123 (Hou et al., 2017) | 89 | 77 | 75 | 81 | **98** | 91 | 76 | 79 | 83 | 92 | **95** | 80 | 87 | 94 | 95 | 96 | 94 | 96 | 85 | 81 | 90 |
| VAE-345 (Hou et al., 2017) | 89 | **80** | 78 | **82** | **98** | **95** | **77** | **81** | 85 | 93 | **95** | 80 | 88 | 94 | **96** | **99** | **95** | **97** | 89 | 85 | 95 |
| VAE-WGAN (ours) | 90 | **80** | 79 | **82** | **98** | **95** | **77** | **81** | 86 | 94 | **95** | **82** | 89 | **95** | **96** | 98 | **95** | **97** | 88 | 85 | 94 |

| Method | Mouth S. O. | Mustache | Narrow Eyes | No Beard | Oval Face | Pale Skin | Pointy Nose | Reced. Hairline | Rosy Cheeks | Sideburns | Smiling | Straight Hair | Wavy Hair | Wear. Earrings | Wear. Hat | Wear. Lipstick | Wear. Necklace | Wear. Necktie | Young | Average |
|---|---|---|---|---|---|---|---|---|---|---|---|---|---|---|---|---|---|---|---|---|
| FaceTracer (Kumar et al., 2008) | 87 | 91 | 82 | 90 | 64 | 83 | 68 | 76 | 84 | 94 | 89 | 63 | 73 | 73 | 89 | 89 | 68 | 86 | 80 | 81.13 |
| PANDA-w (Zhang et al., 2014) | 82 | 83 | 79 | 87 | 62 | 84 | 65 | 82 | 81 | 90 | 89 | 67 | 76 | 72 | 91 | 88 | 67 | 88 | 77 | 79.85 |
| PANDA-l (Zhang et al., 2014) | **93** | 93 | 84 | 93 | 65 | 91 | 71 | 85 | 87 | 93 | **92** | 69 | 77 | 78 | 96 | **93** | 67 | 91 | 84 | 85.43 |
| LNets+ANet (Liu et al., 2015) | 92 | 95 | 81 | **95** | 66 | 91 | 72 | 89 | 90 | **96** | **92** | 73 | 80 | 82 | **99** | **93** | 71 | **93** | **87** | 87.30 |
| VAE-123 (Hou et al., 2017) | 80 | **96** | 89 | 88 | 73 | 96 | 73 | **92** | **94** | 95 | 87 | 79 | 74 | 82 | 96 | 88 | **88** | **93** | 81 | 86.95 |
| VAE-345 (Hou et al., 2017) | 88 | **96** | 89 | 91 | 74 | 96 | 74 | **92** | **94** | **96** | 91 | **80** | 79 | 84 | 98 | 91 | **88** | **93** | 84 | 88.73 |
| VAE-WGAN (ours) | 85 | **96** | 89 | 91 | 74 | **97** | 74 | **92** | **94** | **96** | 91 | **80** | 80 | **85** | **99** | 91 | **88** | **93** | 84 | **88.88** |

*Male*, *Smiling*, *Mustache*. Thus, for a given image with latent vector $z$, we can manipulate the facial attribute with the corresponding attribute vector arithmetically, e.g. $z = z + \alpha z_{smiling}$. Figure 2 (b) shows the results for 6 attributes, i.e., *Bald*, *Black hair*, *Eyeglass*, *Male*, *Smiling*, and *Mustache*. We can see that our method can achieve smooth transitions for different facial attributes, demonstrating that the face attributes can be modeled linearly in the learned latent space.

**Facial attributes prediction.** We further evaluate the quality of the learned latent representations of the VAE by applying them to facial attributes prediction. Like Liu et al. (2015), 20,000 face images in the CelebA dataset (Liu et al., 2015) are used for testing while the remaining are used as training data. We used a multi-level feature extraction strategy to extract robust features that cover different image scales. Specifically, 5 VAE-WGAN models are trained independently, each using a different convolutional layer of the VGGNet for computing the perceptual loss. The latent vectors for all the 5 models are concatenated as the final extracted features which are used to train standard linear SVM classifiers to predict the 40 facial attributes in the dataset. Results are shown in Table 1. It is seen that our method outperforms all the previous methods.

## 5 CONCLUDING REMARKS

In this paper, We propose a more stable architecture and several simple yet effective techniques to incorporate variational autoencoder in the framework of generative adversarial network. Our model can generate more consistent and realistic human faces with clearer noses, eyes, teeth, hair textures as well as reasonable background. In addition, we further study the quality of the learned representation and achieve new state-of-the-art performance for facial attribute prediction.

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
