# OpenReview forum: "Improving Variational Autoencoder with Deep Feature Consistent and Generative Adversarial Training"
_ICLR.cc/2018/Workshop — Reject_

### Official Review · AnonReviewer3 · 2018-03-09

**Rating:** 3
**Confidence:** 5

**Review:**

The paper proposes to improve the visual quality of VAE samples by pairing the model with two additional components: 1) a pre-trained classifier used as a feature extractor to align the reconstructed images with their associated input in a more abstract, “perceptual” feature space, and 2) a discriminator network used to provide an additional adversarial loss signal.

The first issue I have with the proposed approach is that it feels very ad-hoc: several architectural components are stitched together with loose justification. The idea of augmenting VAE training with auxiliary losses is itself not very new: aside from the cited work of Ridgeway et al. (2015) and Hou et al. (2017), work done by Larsen et al. (2016), Lamb et al. (2016), and Dosovitskiy and Brox (2016)  explore very related ideas. It is unclear to me what the proposed approach brings to the table.

The second (and more concerning) issue I have is the apparent lack of awareness of the line of work the authors’ proposed approach inscribes itself in. Even a cursory read of the cited Hou et al. (2017) paper points towards (uncited) prior work which this paper borrows heavily from. To my knowledge, the idea of augmenting the VAE reconstruction loss with auxiliary reconstruction losses derived from a pre-trained classifier was first studied in Dosovitskiy and Brox (2016) and Lamb et al. (2016). Likewise, the idea of augmenting the VAE loss with an adversarial loss was first studied in Larsen et al. (2016) and Dosovitskiy and Brox (2016). Finally, the facial attribute manipulation experiment features a procedure (exemplified by a “smiling vector”) that appears to be lifted directly from White (2016) without attribution.

For these reasons, I don’t think the paper should be accepted.

References:

Larsen, A. B. L., Sønderby, S. K., Larochelle, H., & Winther, O. (2016). Autoencoding beyond pixels using a learned similarity metric. In Proceedings of the International Conference on Machine Learning.

Lamb, A., Dumoulin, V., & Courville, A. (2016). Discriminative regularization for generative models. arXiv preprint arXiv:1602.03220.

Dosovitskiy, A., & Brox, T. (2016). Generating images with perceptual similarity metrics based on deep networks. In Advances in Neural Information Processing Systems.

White, T. (2016). Sampling generative networks: Notes on a few effective techniques. arXiv preprint arXiv:1609.04468.

---

### Official Review · AnonReviewer1 · 2018-03-10
**Quantitative results are ok, qualitative results are unsatisfying**

**Rating:** 7
**Confidence:** 4

**Review:**

This approach uses a GAN-based approach to sample natural faces with a precise attribute exhibited therein.
The idea seems nice, and the quantitative results are convincing. In practice, the generated images are able to trigger an attribute predictor which capture the synthetic modification to the original image as a legit facial attribute.
I'm happy with these results. The bad comes with the qualitative results, where the generated faces in some cases are weird (bald, black hair) while in some other are more impressive (smiling, male-female overall)

---

### Decision · Program_Chairs · 2018-03-20
**ICLR 2018 Workshop Acceptance Decision**

**Decision:**

Reject

**Comment:**

Based on the reviews, this paper has not been accepted for presentation at the ICLR workshop. However, the conversation and updates can continue to appear here on OpenReview.